



# Effects of climate change on the valley glaciers of the Italian Alps

Rossana Serandrei-Barbero[1], Sandra Donnici[2], Stefano Zecchetto[3]

[1]formerly ISMAR - Institute of Marine Sciences, CNR - National Research Council of Italy, Venice, Italy and Comitato Glaciologico Italiano, Torino, Italy

[2]IGG - Institute of Geosciences and Earth Resources, CNR - National Research Council of Italy, Padova, Italy

[3]ISP - Institute of Polar Sciences, CNR - National Research Council of Italy, Padova, Italy and Department of Electrical Engineering, Persian Gulf University, Bushehr, Iran

*Correspondence to*: Sandra Donnici (sandra.donnici@cnr.it)

**Abstract.** The behaviour of the valley glaciers of the Italian Alps as a result of the climate changes expected for the 21st

century has been investigated. From 1980 to 2017 the average length reductions of these glaciers has been 16% and their average areal reduction around 22%, much smaller than the overall glacier retreat of the Alps. Their mean observed shortening was about 500 m for a temperature increase of 1.4 °C. To quantify the valley glacier life expectancy, a model estimating their length variations from the air temperature variations of the EuroCordex climatological projections of six different models under RCP4.5 and RCP8.5 scenarios has been used. The ensemble mean temperatures in the Italian Alps region under these

scenarios indicate increases of temperature of ~2°C and ~4°C from 2018 to 2100 respectively. In both scenarios, the glacier model projections show a constant retreat until the eighties, weakening towards the end of the century. As expected, it resulted more severe under the RCP8.5 (from 22% to 48%) than under the RCP4.5 (from 10% to 25%) scenario, with a mean length shortening of 35% and 13% respectively by 2100. The model used estimates that the majority of the valley glaciers could better resist the climate change.

**1 Introduction**

The fluctuations of glacier fronts are mainly due to their response to year-to-year variability of the air temperature and precipitation. This variability alone can explain kilometric variations in length on a secular scale also in constant climatic conditions (Roe, 2011). The retreat rate of the last decades leaves no doubt about the incidence of the current climate anomaly and focuses interest on the behaviour of glaciers and the incidence of different morphometric factors on their shrinkage (e.g.

Charalampidis et al., 2018). The highly variable response of glaciers to climate changes involves changes in length, area and flow velocity that complicate the assessment of future glacier behaviour (Carturan et al., 2013). The acceleration shown by the glacial retreat in recent decades also raises new questions about the glacier life expectancy also regarding the influence of their geometry on their response to climate variations.

In recent decades, the response of glaciers to the rise in temperature since the 1980s has been analyzed using models that use

the concept of climatic sensitivity (i.e., the relation between glacier length decrease and temperature increase) and attribute to





temperature the role of main forcing (Oerlemans, 2005, 2012). Models frequently use mass balance (Brown et al. 2010; Braithwaite et al. 2013; Christian et al. 2018), but a generalization of the glacier behaviour, and more specifically how the glaciers respond to the climatological variations, seems also obtainable through models dealing with the glacier length variations, given the large amount of data that are collected annually at their terminus, at present available at the World Glacier

Monitoring Service (2017 and earlier reports). For this reason, several studies have analyzed the effects of climate changes on glacier snout fluctuations (Calmanti et al., 2007; Bonanno et al., 2014; Nigrelli et al., 2015; Peano et al., 2016).

The application of the model proposed by Oerlemans (2005) to the Tauern Alps glaciers (Eastern Italian Alps), obtained after a recalibration of the climate sensitivity and response time, verified its ability to reproduce the retreat of the glacier fronts even for the glaciers for which length fluctuations data are not available (Zecchetto et al., 2017). On the Tauern Alps in the last 35

years, the frontal retreat of valley glaciers has been less than that of mountain glaciers and this lower climatic sensitivity is also reflected in the projections provided by the model, which are less dramatic for the valley glaciers with respect to mountain glaciers (Serandrei-Barbero et al., 2019). The present work extends these analyses to all the valley glaciers located on the Italian side of the Alps from 2018 to 2100. It is aimed to study their future behaviour in terms of their length reduction under the increase of air temperature according to mid-range and high-end scenarios, i.e. the Representative Concentration Pathways

(RCPs) RCP4.5 and RCP8.5 scenarios.

The study is structured as follows: Section 2 introduces the glacier model used in this work. Section 3 illustrates the data employed, that is the observed data of the glaciers length and temperature and precipitation since 1980, and the climatological data obtained from Euro-Cordex (Jacob et al., 2014) (https://euro-cordex.net/). Section 4 presents the results, expressed in terms of the observed glaciers retreat from 1980 to 2017 and of climatological projections, then followed by Section 5 devoted

to the results discussion. Section 6 summarizes the results.

## 2 The glacier model

The glaciers of the Italian Alps studied in this work are regularly observed through measurements of their annual snout position: the main parameters describing their morphological characteristics, i.e. the slope and length, are known but other important parameters such their ice thickness are unknown. This limited drastically the choice of the glacier model to use for the

climatological projections: in fact, the available glacier (slope, length and snout fluctuations) and meteorological (precipitation and air temperatures) experimental data led us to address to the model proposed by Oerlemans (2005), simply relating the glacier length fluctuations $L'(t)$ to the air temperature $T'(t)$ fluctuations. Others more sophisticated glacier models, such as that by Roe and O'Neal (2009), cannot be applied to the valley glaciers of the Italian Alps essentially because the ice thickness is not available. The model adopted in this work writes as

$$\frac{dL'(t)}{dt} = -\frac{c_s \, T'(t) + L'(t)}{\tau},$$    (1)



where $C_s$ is the climate sensitivity (m K$^{-1}$) and $\tau$ is the glacier response time (year). $L'(t)$ (m) is the variation in the glacier length with respect to its average value over the period considered. The climate sensitivity $C_s$ is defined as

$$C_s = \frac{\bar{P}_{ann}^{1/2}}{c_1 * s} , \tag{2}$$

variations. where $\bar{P}_{ann}$ (m y$^{-1}$) is the mean annual precipitation at the glacier site and $s$ is the glacier slope. The response time

$\tau$ is

$$\tau = \frac{c_2}{c_3 \bar{P}_{ann}^{1/2} s (1+20s)^{1/2} \tilde{L}^{1/2}} , \tag{3}$$

where the constant $c_3$ = 0.006 was obtained from calibration with numerical simulations (Oerlemans, 2005) and $c_1$ = 0.0078 ± 0.0004 and $c_2$ = 1.35 ± 0.14 result from a re-calibration carried out by Zecchetto et al. (2017) on the Italian Eastern Alps. $\tilde{L}$ is the characteristic glacier length over the considered period.

Both $C_s$ and $\tau$ depend on the morphological characteristics of the glaciers, i.e. glacier length $\tilde{L}$ and slope $s$, and on the mean annual precipitation $\bar{P}_{ann}$. Thus, they are not independent of each other. Under similar morphological characteristics, glaciers located in more rainy locations have larger $C_s$ and smaller $\tau$ than those in dryer areas. On the contrary, in areas of similar precipitation rates, steeper glaciers have smaller $C_s$ and $\tau$.

The model can be used (Oerlemans, 2011; Leclercq and Oerlemans, 2012) if

$$\sigma_L / \tilde{L} \ll 1 , \tag{4}$$

where $\sigma_L$ is the standard deviation of the glacier length and $\tilde{L}$ the characteristic glacier length over the considered period. This condition states that the model can be applied only if the glacier variations are small with respect to its length; thus, implicitly, that the the model does not account for all the non-linear and local factors influencing a glacier's life. For the climatological projections, Eq. 4 has been computed over time windows of 15 years. The results provided in this paper are all compatible

with Eq. 4, and all the conclusions reached must be viewed in light of this constraint.

In the climatological projections spanning long periods, it is unlikely that the values of mean annual precipitation are constant with time; also the representative glacier length $\tilde{L}$ cannot be taken as constant, since the expected glacier reduction due to climate temperature rise.

In this work, we modified the original formulation of Oerlemans (2005) reported in Eq. 1 allowing variations of $\tilde{L}$ and $\bar{P}_{ann}$

with time, impling that $C_s$ and $\tau$ (Eqs. 2 and 3) are not constant any more but time dependent. Therefore, Eq. 1 becomes:

$$\frac{dL'(t)}{dt} = \frac{C_s(t)T'(t) + L'(t)}{\tau(t)} , \tag{5}$$

a first order non-linear differential equation of the type $\quad \frac{dL'(t)}{dt} = F(t, L') ,$

with $\quad F(t, L') = -\frac{C_s(t)T'(t) + L'(t)}{\tau(t)} , \qquad C_s(t) = \frac{\bar{P}_{ann}(t)^{1/2}}{c_1 * s} ,$ and $\quad \tau(t) = \frac{c_2}{c_3 \bar{P}_{ann}(t)^{1/2} s (1+20s)^{1/2} \tilde{L}(t)^{1/2}} ,$



where $\bar{P}_{ann}(t)$ and $\tilde{L}(t)$ are the low frequency climatological precipitation and  representative glacier length computed over a

15 years time window. Tests run considering $C_s$ and $\tau$ constant or slowly variable yielded differences smaller than 5% of the

determination of the glacier length variations.

In the climatological projections, we have used Eq. 5, solved numerically using the well established Runge-Kutta method

*(*Atkinson, 1989).

### 3 The data

The data presented in this section are of two kinds: the observed data, available from 1980 to 2017, which include the glaciers

snout positions and the air temperatures and precipitation at annual frequency, and the climatological model data derived from

the Euro-Cordex projections until 2100.

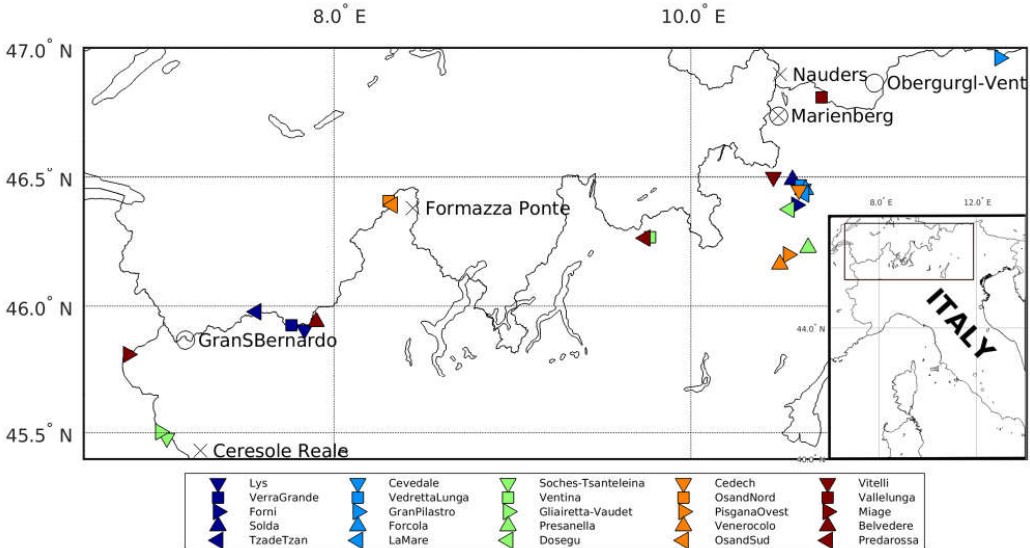

**Figure 1: The position of the glaciers and in-situ climatological stations considered in this work. Circles indicate the stations**
**providing air temperature, crosses the precipitation.**

### 3.1 The glacier data

Given the subjectivity of the primary glacier classification and the possible transition of some glaciers to a different typology

due to their severe recent decline, this study examines only the italian glaciers classified as valley glacier, i.e. glaciers flowing

down a valley and in consequence having a distinct tongue (Cogley et al., 2011). Although some other Italian glaciers are



defined as valley glacier in literature (see for example Bonanno et al., 2014), those classified as valley glaciers in the Italian Glacier Inventory (Smiraglia and Diolaiuti, 2015) represent 3% of the totality, that is 25 glaciers (96.06 km$^2$) compared to a total of 903 glaciers (369.90 km$^2$), 57% being mountain glaciers and 40% glacierets. Valley glaciers are few, but their length variations are measured since long time; the presence of the valley tongue, generally easily accessible, and their frequent large dimensions favoured the choice of them for the annual measurements, i.e the distance year to year between some points of the

glacier terminus and the corresponding fixed landmarks on the ground.

We consider in this work the 25 glaciers reported as valley glaciers by Smiraglia and Diolaiuti (2015), mapped in Fig. 1, which reports their position along with that of the in-situ meteorological stations providing air temperature and precipitation. The area including them is a part of the Alps of about 700 km in longitude and 200 km in latitude.

Among them, three glaciers have been discarded from analysis: the Miage glacier is a debris-covered glacier, the front of

which is inaccessible; the Belvedere glacier between 2001 and 2005 was travelled by a surge, a phenomenon believed to be cyclical  (Van de Wal and Oerlemans, 1995);  the Predarossa glacier is split since 1993 into two sectors at different altitudes and scarcely connected to the sides of a rocky outcrop: an upper-altitude sector seems active as a probable circus glacier, a lower-altitude sector, extended along the slopes on the orographic left, is fed by avalanche and classifiable as slope glacier.

For 19 of the remaining 22 valley glaciers, the annual ground measurements of snout fluctuations collected by the Italian

Glaciological Committee are available since the 70s of the last century, that is, from the beginning of the last positive fluctuation before the current retreat. Through the annual reports of these measurements (CGI, 1981-2018), the length and slope of the considered glaciers were updated to 2017. For four of them, snout measurements stopped before 2017 due to major morphological changes induced by the predominant processes of down-wasting during the early 21st century: on the Tza de Tzan glacier the measurements at the front stopped in 2001 with the emergence of the rocky escarpment which prevents the

feeding of the tongue and converts the valley glacier into a hanging glacier; for the Lys Glacier, snout measurements stopped in 2008 when the valley tongue separated from the feeder basin; on the Pisgana glacier, since 2011 the tongue has been interrupted by a rocky outcrop that limits its feeding; finally, the Verra Grande glacier tongue is not accessible since 2014 for the repositioning of the glacier terminus above a rocky outcrop. The length variations of the three glaciers without measurements of their terminus fluctuations (Osand Sud, Vitelli and Vallelunga glaciers) have been obtained from the historic

observed temperature anomalies through the model illustrated in Section 2.

Table 1 shows the main morphometric parameters of the 22 valley glaciers considered. The glacier length and area data were collected on the ground between the end of the 70s and the early 80s during the campaigns for the compilation of the World Glacier Inventory (WGMS, 2017 and earlier reports) and 1980 constitutes the reference year for the length and area values. The lengths in 2017 are from the annual field measurements of Comitato Glaciologico Italiano (1981-2018) starting from 1980

lengths. The 2015 area values are derived from orthophotos or satellite images taken between 2007 and 2011 (Smiraglia and Diolaiuti, 2015).





Between 1980 and 2017, the 22 considered valley glaciers lost an average length of 542 m and an average area of 0.88 km$^2$ for a total loss of 19.36 km$^2$. In the same period on the Italian side of the Alps the glacierized total area loss was 238.66 km$^2$ corresponding to 39% with respect to the 1980 glacierized total area (Smiraglia and Diolaiuti, 2015).


**Table 1: Main parameters of the 22 valley glaciers considered. Slope is the angle between the glacier surface and the horizon. The values of climate sensitivity $C_s$ and response time $\tau$ were computed by Eqs. 2 and 3.**

| Alpine sector | Glacier | WGI code | latitude N | longitude E | Area 1980 (km$^2$) | Area 2015 (km$^2$) | Δ area (%) | Length 1980 (m) | Length 2017 (m) | Δ length (%) | Slope 1980 (°) | $C_s$ (mK$^{-1}$) | $\tau$ (years) |
|---|---|---|---|---|---|---|---|---|---|---|---|---|---|
| Western Alps | Soches-Tsanteleina | IT4L01514011 | 45° 28' 51" | 7° 03' 41" | 3.32 | 2.77 | -17 | 3400 | 2781 | -18 | 12 | 571.6 | 7.1 |
| | Gliairetta-Vaudet | IT4L01515021 | 45° 30' 17" | 7° 01' 31" | 4.41 | 3.72 | -16 | 3200 | 2810 | -12 | 19 | 333.0 | 3.3 |
| | Tza de Tzan | IT4L01522024 | 45° 58' 40" | 7° 33' 44" | 3.95 | 3.27 | -17 | 3700 | 3600 (2001*) | | 19 | 387.3 | 3.9 |
| | Verra Grande | IT4L01504004 | 45° 55' 34" | 7° 45' 33" | 7.28 | 6.6 | -9 | 5100 | 4425 (2014*) | | 18 | 421.8 | 3.6 |
| | Lys | IT4L01502002 | 45° 54' 31" | 7° 49' 58" | 11.82 | 9.89 | -16 | 5600 | 5357 (2008*) | | 20 | 319.1 | 2.4 |
| | Osand Sud | IT4L01216015 | 46° 23' 34" | 8° 19' 39" | 3.64 | 2.21 | -39 | 2810 | 1712** | -39** | 15 | 491.1 | 6.1 |
| | Osand Nord | IT4L01216018 | 46° 24' 29" | 8° 18' 20" | 1.98 | 1.31 | -34 | 2878 | 2703 | -6 | 14 | 618.6 | 8.5 |
| Eastern Alps | Ventina | IT4L01122009 | 46° 16' 02" | 9° 46' 17" | 2.26 | 1.89 | -16 | 3171 | 2684 | -15 | 23 | 291.5 | 3.1 |
| | Vitelli | IT4L01139003 | 46° 30' 03" | 10° 27' 39" | 1.82 | 1.89 | 4 | 3100 | 2174** | -30** | 16 | 442.7 | 5.5 |
| | Cedech | IT4L01137018 | 46° 27' 00" | 10° 36' 22" | 2.84 | 2.07 | -27 | 3085 | 2565 | -17 | 20 | 304.5 | 3.4 |
| | Forni | IT4L01137024 | 46° 23' 32" | 10° 35' 28" | 13.24 | 11.34 | -14 | 5062 | 4363 | -14 | 15 | 449.5 | 4.7 |
| | Dosegù | IT4L01137031 | 46° 22' 34" | 10° 33' 07" | 3.44 | 2.16 | -37 | 3149 | 2606 | -17 | 14 | 521.8 | 7.2 |
| | Pisgana Ovest | IT4L01028006 | 46° 11' 59" | 10° 32' 40" | 2.47 | 2.49 | 1 | 2650 | 2190 (2010*) | | 15 | 415.5 | 5.8 |
| | Venerocolo | IT4L01028011 | 46° 09' 46" | 10° 29' 56" | 1.18 | 0.82 | -31 | 2144 | 1968 | -8 | 17 | 544.9 | 9.2 |
| | Presanella | IT4L00102413 | 46° 13' 35" | 10° 39' 24" | 3.92 | 2.79 | -29 | 3200 | 3093 (2007) | | 18 | 330.9 | 3.8 |
| | La Mare | IT4L00102516 | 46° 26' 05" | 10° 38' 00" | 4.75 | 3.53 | -26 | 3470 | 2982 | -14 | 24 | 337.3 | 3.6 |
| | Forcola | IT4L00112125 | 46° 27' 07" | 10° 38' 26" | 2.52 | 1.77 | -30 | 3500 | 2596 | -26 | 19 | 361.0 | 4.0 |
| | Cevedale | IT4L00112126 | 46° 27' 20" | 10° 37' 45" | 3.20 | -- | -- | 3700 | 2841 | -23 | 17 | 411.7 | 4.7 |
| | Vedretta Lunga | IT4L00112128 | 46° 28' 05" | 10° 36' 55" | 3.22 | 1.76 | -45 | 3600 | 2680 | -26 | 13 | 537.0 | 7.0 |
| | Solda | IT4L00112417 | 46° 29' 23" | 10° 34' 05" | 6.48 | 5.5 | -15 | 4200 | 4020 | -4 | 21 | 346.7 | 3.5 |
| | Vallelunga | IT4L00112907 | 46° 48' 41" | 10° 43' 57" | 8.55 | 7.35 | -14 | 3900 | 3180** | -18** | 20 | 344.7 | 3.5 |
| | Gran Pilastro | IT4L00121313 | 46° 57' 51" | 11° 43' 45" | 2.62 | 1.73 | -34 | 3700 | 3060 | -17 | 14 | 402.2 | 6.5 |
| | **mean values** | | | | **4.54** | **3.66** | **-22** | **3560** | **3018** | **-16** | **17** | **417.5** | **5.0** |

\* glaciers not considered in the projections
\*\* unmeasured glacier

### 3.2 The meteorological data

The air temperature and total precipitation used in this work are both historical and forecast data. The former, obtained from the HIStorical instrumental climatology surface time series of the greater ALPine region (HISTALP, www.zamg.ac.at/histalp/, Auer et al., 2007), are available from 1980 to 2014. The stations selected are positioned between 1300 m and 2470 m of altitude: their location is reported in Fig. 1, with circles and crosses.

Figure 2 reports the historical time series of temperature fluctuations (left panel) and annual total precipitation (right panel)
derived from the data of the available stations; the latter sorted for their longitudinal location. The temperature fluctuations, i.e. the temperature values with respect to their mean, exhibit a similar trend, an increase of about 1.4° C in the period 1980-2015, while the total precipitation show no trends but larger values in the western than in the eastern part of the region under study. Total precipitation data have been used to compute $C_s$ and $\tau$ (see Section 2, Eqs. 2 and 3) for all the glaciers. The air





temperature variations have been used instead to reproduce the length historic variations of the three glaciers not measured
(Osand Sud, Vitelli and Vallelunga glaciers).

The atmospheric climatological model data used are the Euro-Cordex (Jacob et al., 2014) (https://euro-cordex.net/) temperature

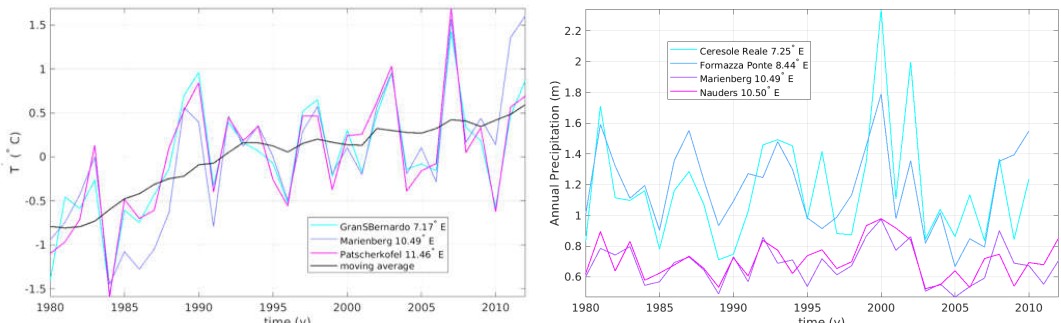

**Figure 2: Time series of experimental air temperature variations (left panel) and annual total precipitation (right panel) from the in-situ stations available.**

and precipitation projections of the climatological Representative Concentration Pathways (RCPs) RCP4.5 and RCP8.5 scenarios. The Euro-Cordex are regional climate model simulations for the European domain with resolution of 0.11 degree (EUR-11, ~12.5km). The two climatological scenarios are based on greenhouse gas emission corresponding to stabilization of
radiative forcing after the 21st century at 4.5 W/m$^2$ (RCP4.5), rising radiative forcing crossing 8.5 W/m$^2$ at the end of 21st century (RCP8.5) (Moss et al., 2010; Nakićenović et al., 2000; Van Vuuren et al., 2008). Six different regional climatological models providing air temperature and total precipitation, i.e. CCLM4-8-17.v1 (Rockel et al., 2008), HIRHAM5.v3 (Christensen et al., 1998), Racmo2.2 (Van Meijgaard et al., 2012), RCA4.v1 (Samuelsson et al., 2011), WRF (Skamarock et al., 2008), and REMO2009 (Jacob et al., 2012), have been used in order to derive ensemble averages of the glacier future
length variations. In the following, we will refer as ensemble averages the mean of the time series from the different climatological models.

The ensemble air temperature anomalies at the glaciers position are reported in Fig. 3 for the two scenarios considered. While for the milder RCP4.5 there will be a rise of temperature of about 1.6°C from 2020 to 2086 and then a light decrease to 2100, RCP8.5 indicates an increase of about 4.5°C from 2020 to 2095, followed by a light decrease of some tenths of a degree. The
mean increase of temperature is 0.020 ± 0.014 °C/y and 0.053 ±0.017 °C/y respectively.

In some period, the air temperature variations $T'$ vary significantly (~ 0.3 °C) according to the glacier size, as in the period 2020-2040 and 2080-2100. This is because the simulations have been carried out using the climatological temperature and precipitation closest to the glacier location.

## 4 Results

### 4.1 Glaciers retreat until 2017

Between 1980 and 2017 the loss of length of the valley glaciers was between 4% and 26% with an average contraction of 16% (about 500 m). The areal reduction was between 9% and 45% with an average shrinkage around 22%. Table 1 provides an overview of the length and area variations that occurred between 1980 and 2017 together with the $C_s$ and $\tau$ values obtained from Eqs. 2 and 3. The climate sensitivity $C_s$ ranges between 291 m K$^{-1}$ (Ventina Glacier) and 619 m K$^{-1}$ (Osand Nord Glacier) with a mean value of 417 m K$^{-1}$. A temperature increase of 1.4 °C from 1980 to 2015 produces a mean length decrease of 584 m, consistent with the observed average shortening of 542 m. The glacier response time $\tau$ is between 2 and 10 years, with a mean value of 5 years.

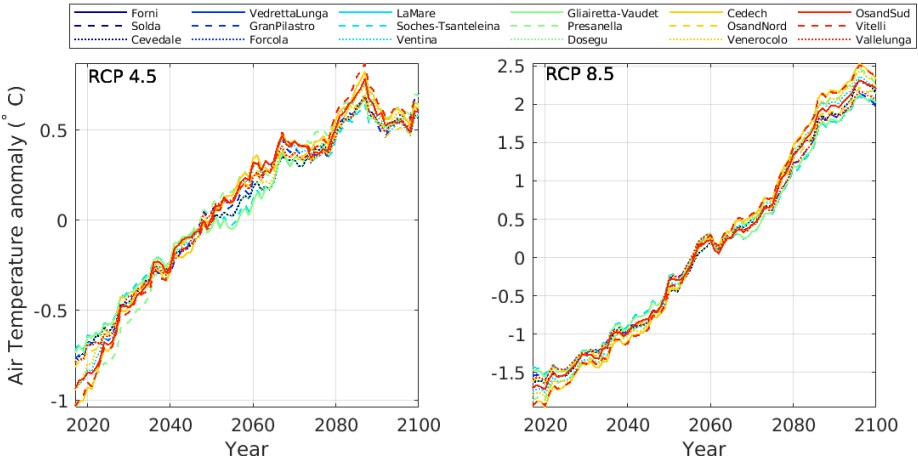

**Figure 3: The ensemble air temperatures variations over the six regional models at the grid points closest the glaciers position from 2018 to 2100. Left panel: RCP4.5 scenario. Right panel: RCP8.5 scenario. The colours are sorted from the longest (blue) to the shortest (red) glacier.**

The climate sensitivity $C_s$ (Eq. 2) depends on the glacier slope and the total annual precipitation. Glaciers with slope 18° < s < 27° have $C_s$ < 350 m K$^{-1}$ and response times 1.7 < $\tau$ < 3.8 years; glaciers with a more gentle slope 12° < s <16° have $C_s$ > 450 m K$^{-1}$ and $\tau$ of 7 - 9 years. Figure 4 reports $Cs$ versus the glacier slope $s$ for the 22 valley glaciers considered in this work: as expected by Eq. 2, $Cs$ decreases, as s increases, in average – 24 m for one degree of slope.





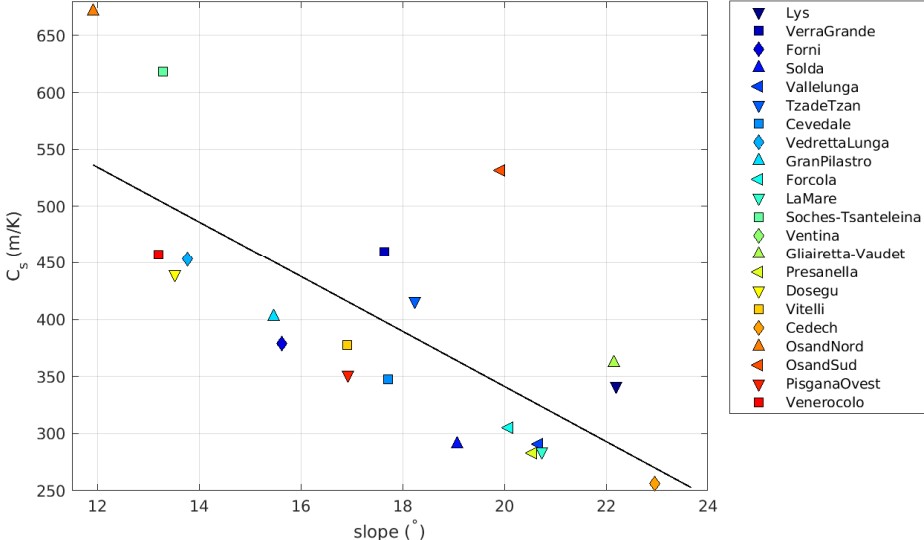

**Figure 4: $C_s$ versus the glaciers slope s for the 22 valley glaciers considered in this work. The solid line is the linear best fit. The glaciers in the legend are sorted from the longest (blue) to the shortest (red).**

Figure 5 reports the observed retreat from 1980 to 2017 of the measured glaciers: all exhibit a continuous shrinkage since
1990, but with very different rates, from ~7% of the Solda and Osand Nord glaciers (respectively among the longest and the
shortest) to ~25% of Forni and Forcola glaciers.

### 4.2 Glaciers projections to 2100

Climatological glacier lengths have been obtained forcing the glacier model (Eq. 5) with the annual air temperature variations
and the total precipitation derived by each of the six atmospheric climatological models chosen (see Section 3.2) at the grid
point closest to the glacier location.

The results are shown as ensemble means, derived averaging the glacier lengths variations obtained from each atmospheric
climatological model. The Tza de Tzan, Verra Grande, Lys, and Pisgana W glaciers, that before 2017 underwent morphological
variations, cannot be anymore considered valley glaciers. Therefore, they were not considered in the projections, which include
instead the three unmeasured glaciers (Osand Sud, Vitelli and Vallelunga), the 2017 length of which derived by the glacier
model simulations. Consequently, the climatological projections have been carried out over 18 glaciers.

Figure 6 reports the ensemble glacier length variations in percent from 2018 to 2100 under the RCP4.5 (top panel) and RCP8.5
(bottom panel) scenarios. These have been obtained averaging the projections derived by every single climatological model

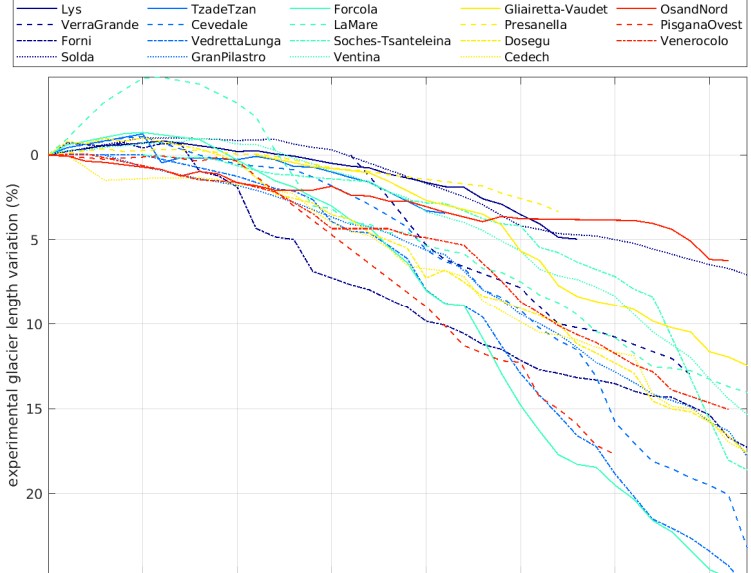

**Figure 6: The percent glaciers length variations derived from climatological projections from 2018 to 2100.
Top panel: RCP4.5 scenario. Bottom panel: RCP8.5 scenario. The colours are sorted from the longest (blue)
to the shortest (red) glacier.**





respecting the glacier model constraint of Eq. 4, computed over time windows of 15 years, set as $\sigma_L/\tilde{L} < 0.1$, i.e. glacier variations at least one order of magnitude smaller than the glacier length.

Figure 7 shows the ensemble values of $\sigma_L/\tilde{L}$ : for RCP4.5 scenario (top panel) $\sigma_L/\tilde{L} < 0.05$, thus respecting the constraints of Eq. 4 for the whole projections period; for scenario RCP8.5 (bottom panel) $\sigma_L/\tilde{L}$ is greater than 0.1 in the late seventies for two glaciers (Dosegu and Venerocolo), which therefore cannot be projected until 2100. These glaciers will probably disappear within the end of the century.

In both scenarios, there is a constant retreat until the eighties of this century, weakening towards the end of the century. As

expected, it is more severe under the RCP8.5 (from 22% to 48%) than under the RCP4.5 (from 10% to 25%), since the almost double temperature rise foreseen by the farther scenario (see Fig. 3).



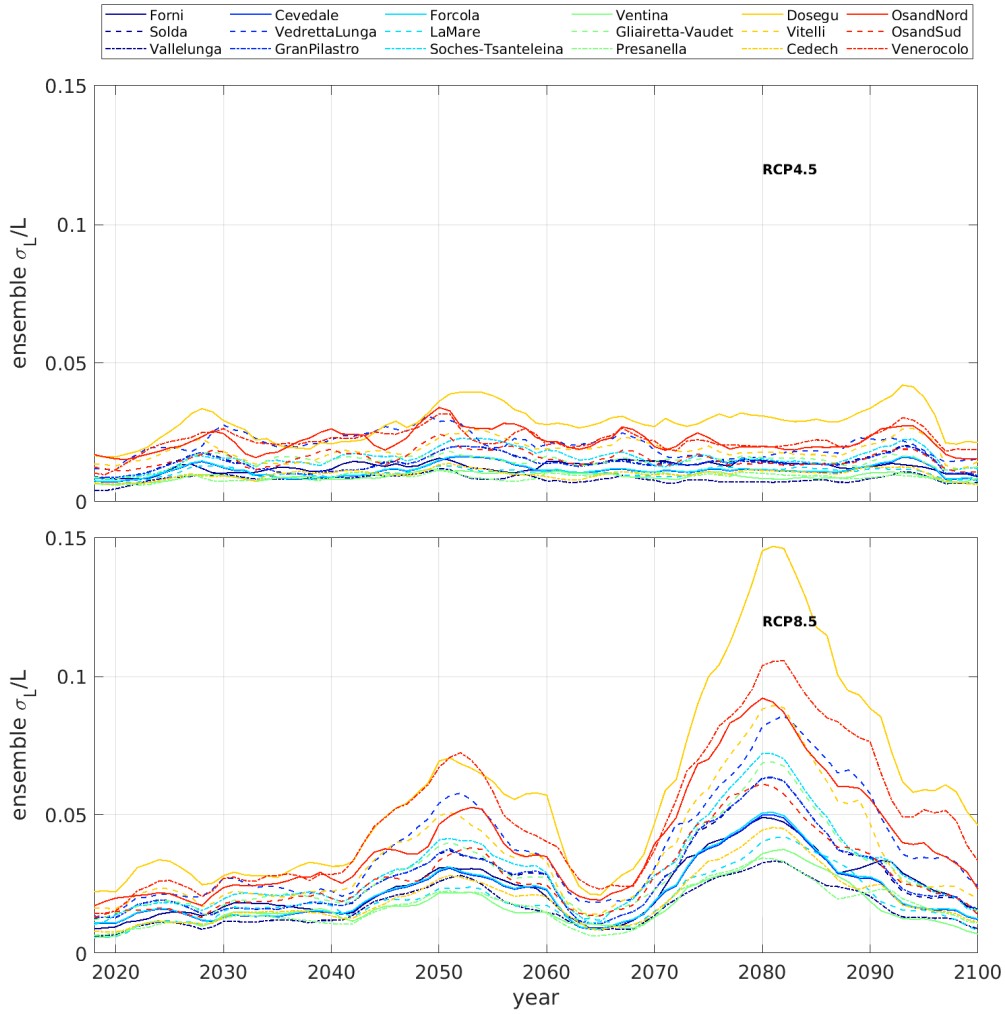

**Figure 7: Ensemble values of $\sigma_L/\tilde{L}$ derived from climatological projections from 2018 to 2100. Top panel: RCP4.5 scenario. Bottom panel: RCP8.5 scenario. The colours are sorted from the longest (blue) to the shortest (red) glacier.**

Figure 8 shows the mean percent of the glacier length reductions shown in Fig. 6: by 2100 there will be a loss of 35% and 13% of the glacier lengths under RCP8.5 and RCP4.5 respectively. The error bars, accounting both for the different behaviour of the glaciers under the same climatological forcing and for the different meteorological forcing provided by the six different

climatological model used, are always smaller than ±7%, making reliable the figure of the mean retreat. Therefore the different climatological models impact on the glacier model results for less than 10%.

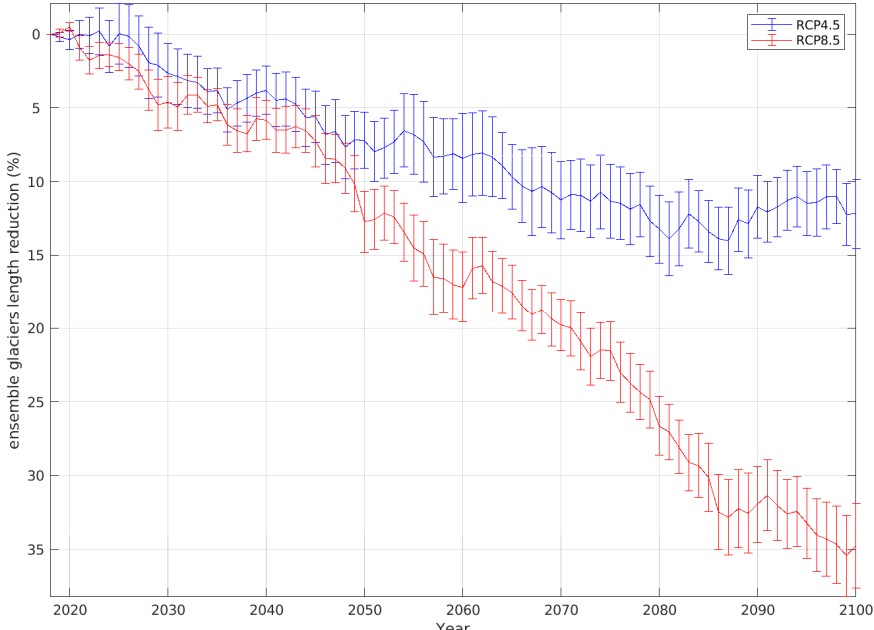


**Figure 8: The mean percent glacier length reduction from 2018 to 2100 according to the RCP4.5 and RCP8.5 climatological scenarios.**

These results indicate that, even under the most severe RCP8.5 scenario, the majority of glacier length variations will not be so large as to menace the existence of the glaciers to 2100 with the exceptions of two (Venerocolo and Dosegu) , which will

preserve the present behaviour characterized by glacier size much bigger than their annual variations. This even if some of them, i.e. Vitelli, Osand Nord and Vedretta Lunga glaciers, will almost half their 2018 length, in accordance with the general trend that attributes greater sensitivity to gently sloping glaciers (Fig. 4).



## 5 Discussion

Studies dealing with alpine glaciers generally group them based on their geographic location (Carturan et al., 2013; Fischer et
al., 2015 among others) or their size (DeBeer and Sharp, 2009; Huss and Fischer, 2016) while their primary classification is
not considered, which seems to have some influence on the climatic sensitivity.

On the valley glaciers, from 1980 to 2017 the mean length loss is 16%. Their average areal shrinkage of 22% between 1980
and 2015 (Table 1) shows their smaller retreat with respect to the general shrinkage in the European Alps estimated around
40% as a lower bound (Knoll and Kerschner, 2009; Lambrecht and Kuhn, 2007; Nigrelli et al., 2015; Paul, 2002; Zemp et al.,
2006, 2015).

On the Eastern Alps, a previous comparison between valley and mountain glaciers belonging to the same size classes
highlighted the lower retreat of valley glaciers (Serandrei-Barbero et al. 2019).

All the 22 valley glaciers considered in this paper have an area > 1 km$^2$, except one, while 90% of the glaciers on the Italian
Alps (812 glaciers) have areas < 1 km$^2$. In general larger glaciers show regresses inversely proportional to their size (Serandrei-
Barbero et al. 1999; Paul et al. 2004, 2011; Diolaiuti et al. 2012; Carturan et al. 2013); this could significantly contribute to
their expected minor retreat. However, on the valley glaciers considered here it does not seem to be any relationship between
size and frontal retreat as it can be inferred from Fig. 5, which reports past percent retreats sorted according to the glacier size.
The frontal variations, as reported in Section 4.1, result influenced by the slope, in accordance with Hoelzle et al. (2003) and
Huss and Fischer (2016), the latter describing values of Cs and $\tau$ generally higher on gently sloping glaciers. The longer
response time of gently sloping glaciers, slowing down the feeding of the valley tongue, would favour processes of nonlinear
decay. On the contrary, the steepness of the glacial mass, which is accompanied by a shorter response time, seems to favour
the feeding of the tongue, leaving rise to lower Cs and minor withdrawals. The response time $\tau$ obtained, between 2 and 10
years, agree with $\tau$ values of 2-4 years of the Kesselwandferner glacier and the Palù glacier on the northern side of the Eastern
Alps (Oerlemans 2007) and, more generally, with $\tau$ values of 2-21 years of the glaciers of the Southern side of the Eastern
Alps (Serandrei-Barbero et al., 2019) of length comparable with the glaciers considered here. However, the response times of
valley glaciers are small compared to the response times between 10 and 20 years generally observed in the Alps (Paul et al.
2004).

The results of the glaciers shrinkage due to the simulated changes of air temperature and total precipitation as foreseen by six
atmospheric models under the RCP4.5 and RCP8.5 climatological scenarios are slightly modulated (2%) by the variations of
the climatological total precipitation which enters in the definition of the glacier sensitivity Cs and response time $\tau$.

The main question concerns the reliability of these estimates. A partial answer to this crucial issue may be derived looking to
the observed retreats shown in Fig. 5 and comparing them with those from the two climatological scenarios in terms of retreat
velocity. The mean retreat velocity from 1980 to 2017 was 15±7 m/y (with a mean length loss of about 500 m), while that of
the climatological scenarios RCP4.5 and RCP8.5, computed over period of the same temperature increase occurred from 1980





to 2017 (~1.4°C), is 10±3 m/y and 15±5 m/y respectively. The glacier retreat speed under the RCP8.5 scenario is very close
       to that observed in the period 1980-2017.

       As for the observed glacier length retreats (Fig. 5), the climatological results reported in Fig. 6 show that glaciers behave
       differently under the same climatological forcing: under the RCP4.5 scenario, there is up to 20% of difference between the
       glaciers with smaller reduction and those of largest percent retreat, which almost doubles under RCP8.5. These differences are

possibly due to the values of Cs and τ of each glacier and, in consequence, to their characteristic length and slope, as well as
       to the annual total precipitation in the glacier site. But other variables could influence the glacier behaviour mainly due to
       differences in the geometric characteristics and topographic features of individual glaciers (Paul et al. 2004; Haeberli et al.
       2007; Oerlemans 2012; Huss and Fischer 2016 among others). Furthermore, the model cannot take into account morphometric
       variations such as changes in the elevation range related to frontal retreat or variations in glacier hypsometry (Kuhn 1985; Paul

and Haeberli 2008; Fischer et al. 2015; Zemp et al. 2015; Charalampidis et al. 2018). The results of the model should be
       considered as first order estimates, even because it cannot account for the changes introduced by positive feedbacks, such as
       the decrease in albedo and the thermal emission from increasing rocky outcrops, as well as possible negative feedbacks, such
       as the contribution of avalanches or the increasing debris cover or shaded area. However, since the majority of the glacier
       simulated  projections satisfies the constraint that the glacier variations are much smaller than the glacier length (Eq.4), thus

longer preserving the present glacier morphology, the results of model can be considered reliable: for the valley glaciers the
       average length loss expected for 2100 under the most severe scenario RCP8.5 is between 13% and 60%, with a mean value of
       35%. The projections obtained under RCP4.5, more favourable, show instead length reduction between 5% and 23%, with a
       mean value of 13%.

       According to the estimate based on the scaling relationship between glacier area and length derived for the mid-latitudes glacier

regions (Machguth and Huss 2014), the observed reduction in length of 16% between 1980 and 2017, corresponds to an area
       loss of 22%. A rough estimate based on the same proportion between length and area decrease indicates for 2100 an area loss
       of about 50% (1.83 km$^2$) for an average length loss of 35% under RCP8.5 scenario; under the RCP4.5 scenario, a length retreat
       of 13% corresponds to an area loss of 20% (0.73 km$^2$).

       On the glaciers of the European Alps, the projections for the end of the century indicate volume losses between 75% and 89%

under RCP4.5 and between 90% and 98% under RCP8.5 (Marzeion et al. 2012; Radic et al. 2014; Huss and Hock 2015;
       Zekollary et al. 2019) with the possible disappearance between 69% to 92% of all the glaciers in the Alps (Zebre et al. 2020).
       Both mountain and valley glaciers are included in this severe estimate, but the behaviour of the few valley glaciers is
       undetectable from that of the great majority of alpine glaciers.

       A comparison is instead possible with the projections of some glaciers on the southern side of the Alps: on the Italian Western

Alps, on 14 glaciers considered as valley glaciers by Bonanno et al. (2014), the mean length loss projected to 2050 is quantified
       in 300-400 m compared to the 2010 length with the RCP4.5 scenario, with larger retreats under RCP8.5 scenario  Even though
       these glaciers are smaller (mean area 2.79 km$^2$) than the glaciers considered in the present study, their projected retreat is
       consistent with the average retreat of 7% (about 200 m) and 13% (about 400 m), expected for valley glaciers in 2050 under





RCP4.5 and RCP8.5 respectively (Fig. 8). Moreover, based on A1B scenario (Nakicenovic et al. 2000), also on the Eastern
Alps the expected length loss in 2100 between 20% and 35% on the valley glaciers is much less than the mean shortening
between 35% and 60% expected on mountain glaciers of the same size class (Serandrei-Barbero et al. 2019). The smaller
decline of the valley glaciers with respect to the totality of Alpine glaciers is also confirmed by the comparison with the
projections of Peano et al. (2016): on the studied mountain glaciers in the Italian Western Alps the expected average length
shrinkage in 2100 under the future scenarios RCP4.5 and RCP8.5 is respectively 47% and 68.5% and the length would become
zero in 2100 for some of these mountain glaciers.

On both valley glaciers and mountain glaciers, length reductions are expected also in case of an attenuation or inversion of the
warming in progress because of the response time τ, due to the imbalance always present between the transient length and the
length in conditions of reached equilibrium (Christian et al. 2018). Citterio et al. (2007) suggest that, in the coming decades,
only the few largest glaciers will be in a condition to survive. The last 35 years glacier behaviour shown in the present study
indicates a minor sensitivity of the valley glaciers compared both to mountain glaciers of Eastern Alps, analyzed with the same
model (Serandrei-Barbero et al., 2019), and to the generality of alpine glaciers, as reported in literature. Therefore, the glacier
size and typology could both play a decisive role in the behaviour of valley glaciers under the climate change, but their
projections obtained from the model suggest, for valley glaciers, a possible less dramatic trend than expected.

**6 Conclusions**

Although the literature indicates that the majority of glaciers will disappear in the coming decades, the valley glaciers seems
able to better resist the changing climate and the majority of them could be still present up the end of 21st century.

Between 1980 and 2017 the valley glaciers of the Italian Alps lost on average 16% of their length and 22% of the surface, a
much lower value than the general retreat of the glaciers of the Alps. By 2100, the model projections under the RCP4.5 and
RCP 8.5 climatological scenarios indicate for these glaciers a mean shortening of 13% and 35% respectively, a much lower
value than the expected retreat of the majority of glaciers in the Alps, as reported in literature.

The differences of observed length reductions are up to 20% in the last 40 years. In the projections, for the next 40 years, the
differences are up to 18% for the scenario RCP8.5 (Fig. 6), consistent with those observed.

Summarizing our results: a) under the RPC8.5 scenario, the mean retreat velocity of the glaciers is similar to that of the period
1980 to 2017 obtained from the observed data; b) given the form of the model used, the main forcing is the air temperature
variation, while total precipitation changes have a weak impact on the results; c) the impact of different climatological models
on the simulated results is less than 10%; d) according to the model, under the RCP8.5 scenario the majority of the valley
glaciers (about 80%) will resist the climate change experiencing retreats less than 50% of their 2017 length and thus probably
maintaining their characteristics of valley glaciers.





The simulated glacier retreats have been derived by a glacier model that does not account for possible non-linear effects and consequent geometric changes. Furthermore, possible disintegration and down wasting could deliver some of the valley glaciers to the typology of mountain glaciers, threatened by a much more pronounced retreat. Moreover, progressive glacier shrinkage and fragmentation will lead to a general increasing glacier melt even under the same climatic conditions.

Despite the effect of climate change indicates on the Alps a severe ice masses decline, due to the minor retreat of the valley

glaciers in the last 35 years with respect to the generality of the alpine glaciers, the model suggests that the valley glaciers are less sensitive to the air temperature and precipitation. Their limited length losses result in a slow glacier retreat, thus longer preserving the glacier typology and making the valley glaciers probably better resist the climate change.

**Conflicts of interest/Competing interests (include appropriate disclosures):** The authors declare that they have no known

competing financial interests or personal relationships that could have appeared to influence the work reported in this paper.

**Code availability (software application or custom code)** Not applicable

**Authors' contributions:** R.S.B and. S.D. collected and analyzed the glaciers data, interpreted and discussed the results and contributed to the paper final version; S.Z. collected and analyzed the climatological data, developed the glaciers model and contributed to the paper final version.

**Acknowledgements**

The field measurements from 1980 to 2017 were supported by the Comitato Glaciologico Italiano: thanks are due to the voluntary operators for annually monitoring glacier snout fluctuations. Authors are grateful to Giovanni Mortara and Stefano Perona (Comitato Glaciologico Italiano), Michela Munari (Provincia Autonoma di Bolzano-Ufficio Idrografico) and Franco Secchieri for providing them with the original data collected on the ground during the campaigns for the compilation of the

World Glacier Inventory. Thanks to Marco Zecchetto (Instituto Superior Técnico, Universidade de Lisboa, Portugal) for the suggestions provided to solve the model equation. Euro-Cordex climatological data have been downloaded from esgf-data.dkrz.de/projects/cordex-dkrz.

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






**Captions**

Figure 1: The position of the glaciers and in-situ climatological stations considered in this work. Circles indicate the stations providing air temperature, crosses the precipitation.

Figure 2: Time series of experimental air temperature variations (left panel) and annual total precipitation (right panel) from 500 the in-situ stations available.

Figure 3: The ensemble air temperatures variations over the six regional models at the grid points closest the glaciers position from 2018 to 2100. Left panel: RCP4.5 scenario. Right panel: RCP8.5 scenario. The colours are sorted from the longest (blue) to the shortest (red) glacier.

Figure 4: $C_s$ versus the glaciers slope s for the 22 valley glaciers considered in this work. The solid line is the linear best fit. 505 The glaciers in the legend are sorted from the longest (blue) to the shortest (red).

Figure 5. Observed glacier length variation respect to 1980 length. The colours are sorted from the longest (blue) to the shortest (red) glacier.

Figure 6. The percent glaciers length variations derived from climatological projections from 2018 to 2100. Top panel: RCP4.5 scenario. Bottom panel: RCP8.5 scenario. The colours are sorted from the longest (blue) to the shortest (red) glacier.

Figura 7. Ensemble values of $\sigma_L/L$ derived from climatological projections from 2018 to 2100. Top panel: RCP4.5 scenario. Bottom panel: RCP8.5 scenario. The colours are sorted from the longest (blue) to the shortest (red) glacier.

Figure 8: The mean percent glacier length reduction from 2018 to 2100 according to the RCP4.5 and RCP8.5 climatological scenarios.

Table 1. Main parameters of the 22 valley glaciers considered. Slope is the angle between the glacier surface and the horizon. 515 The values of climate sensitivity Cs and response time τ were computed by Eqs. 2 and 3.