# Peer review of "Effects of climate change on the valley glaciers of the Italian Alps"

_The Cryosphere, 2021_

## Author Comment (AC2)

**Review** to Serandrei-Barbero et al. (2021): "Effects of climate change on the valley glaciers of the Italian Alps" submitted to The Cryosphere.

The authors present a study of the future glacier length evolution of the Italian valley glaciers feeding a simple glacier model with future climate projections taken from the EURO-CORDEX initiative for RCP4.5 and RCP8.5 scenarios. The main result of the study is that until the end of the century the Italian valley glaciers (representing roughly 26% of the total Italian glacier area) will preserve about 50% of their length of 2017 and thus, their retreat will be slower than other glaciers in the Alps.

This is in agreement with what measured on the ground: on the valley glaciers here considered, from 1980 to 2017 the mean length loss is 16%. Their average areal shrinkage of 22% between 1980 and 2015 (Table 1) shows their smaller retreat with respect to the general shrinkage in the European Alps estimated around 40% as a lower bound.

The lower retreat of valley glaciers was also highlighted by considering together valley and mountain glaciers small to medium-sized (Serandrei-Barbero et al., 2019).

However, as the valley glaciers considered here are medium to large in size, it is possible that this may contribute to their lower retreat (at lines 243-247 of the text: In general larger glaciers show regresses inversely proportional to their size and this could significantly contribute to their expected minor retreat).

We include this consideration also in the conclusions.

We realize that the reviewer does not agree with our results. We are also sorry that the reasons for these conclusions are based on some misunderstandings of the text. Our results do not contradict the general belief that climate change will destroy all the glaciers, but rather indicate a longer survival of valley glaciers than mountain glaciers (and this different behavior has already been demonstrated by field monitoring performed in the past decades).

We are aware that the model used in the work is by far simpler than the models quoted here, but we doubt that projections like those reported in this paper may be reached using other models requiring data not available for the Italian Alps.

Overall, I rate this manuscript not ready for publication and due to the sum of inconsistencies I even suggest a rejection. My main points of concern are:

Missing scientific rigor: The introduction (L21) starts with a misconception. Glacier fluctuations are not a result of air temperature and precipitation variability, they are a result of complex climate-glacier interactions.

Of course, anyone dealing with glaciers is aware of the complexity of climate-glacier interactions (at l. 75 of the manuscript: *thus, implicitly, that the model does not account for all the non-linear and local factors influencing a glacier's life*). In the text, we wrote that the air temperature and the variability of precipitation are the **main parameters** that influence the fluctuations of glaciers, but not the only ones. Furthermore, the importance of temperature is well known (Leclercq&Oerlemans 2012), since the past fluctuations of glaciers, used as a proxy of temperature, reproduce very well the instrumental record of the last century.

We just prefer to make glacier fluctuations a proxy of air temperature and precipitation, because we usually have long-term observations of or established scaling functions for them. Hence, simple glacier models were established for conceptual understanding. However, we must be careful when

interpreting (putative) results of (highly) parametrized processes or inverting them. Fig. 4 is an example of such a putative result. The correlation between the slope and the climate sensitivity is not a result of the study, it is a result of the model design. Because the simplified model defines the climate sensitivity as a function of the slope (Oerlemans, 2001), we detect it as correlation in the data. Cause and effect must not be interchanged.

We did not say in any part of the paper that Fig. 4 is a result of the study, and we believe this is an inference of the reviewer that the text does not support. As in the text, *The climate sensitivity Cs (Eq. 2) depends on the glacier slope and the total annual precipitation* and therefore Fig. 4 is just Eq. 2 applied to our glaciers, which provides an estimate of the rate of change of Cs with slope.

Additionally, the chosen method seems not to be state of the art anymore. Meanwhile ice thickness estimates are available (Farinotti et al., 2017) opening the path to models deriving glacier volume changes (e.g. Maussion et al., 2019), having a higher significance than glacier length changes.

You are right. With respect to the models of new generation, our approach does not represent the status of the art. But this does not mean that more complex models would provide better projections, also given the uncertainties on the several parameters needed to run them. The ground data contains many gaps and inconsistencies and, despite the efforts of the Glacier Monitoring Service, not all glacier outlines are available. Our aim was to use available and verifiable data so that we could have a reliable database: the glacier lengths fulfil this premise and are available on almost all Italian valley glaciers.

We include part of these consideration in the Introduction, where Farinotti et al. (2017) and Maussion et al., (2019) are cited.

Missing model calibration: A previous study of the same authors (Zecchetto et al., 2017) calibrated an existing method of glacier length change modelling (Oerlemans, 2005) on smaller glaciers in the Italian Alps for air temperature reconstructions.

Now, the author team applies the same method without further amendments on the larger Italian valley glaciers. While the model is calibrated on shorter and steeper glaciers,

As the reviewer probably knows, model calibration is a statistical procedure, which minimizes the differences between the model and the data through coefficients. In our case, the model (eq. 1) was compared with the data of 3 glaciers, and the coefficients $c_1$ and $c_2$ of Eqs. 2 and 3 were estimated as $c_1 = 0.0078 \pm 0.0004$ and $c_2 = 1.35 \pm 0.14$ by means of least-squares regression of the function. These values, of course, are mean values and satisfy differently the glaciers, but we cannot have N coefficients for N glaciers.

The glaciers used for calibrations are on average smaller (from 1060 m to 2192 m) than those of the present work (from 1712 m to 5357 m), but not steeper.

it is applied on longer and flatter glaciers, although the authors would have all the data to calibrate the model on the valley glaciers, too. The missing model calibration might explain the low climate

Model calibration is not missing and we wonder how the reviewer could expect larger values of Cs and $\tau$ and why. Our new calibration in Zecchetto et al, 2017 has been shown to work on glaciers longer than those used for calibration (2880 m, 1267 m, 1933 m). This was done because the Oerlemans 2005 calibration did not suit with the glaciers of our region and cannot be used for them.

sensitivities and short response times compared to the original model publication (Oerlemans, 2005) and definitely impacts the results of the study and the conclusions the authors draw.

Oerlemans, 2005 studied 169 glaciers over the world with lengths from 0.3 km to 45 km, while Leclercq&Oerlemens (2012) used 309 glaciers for temperature reconstruction. As far as we know, their coefficients of calibration (0.00204 for Cs and 19.4 for $\tau$) were obtained from fourteen glaciers and then used for all the glaciers. Our procedure was quite similar to that of these quoted works, but with a smaller glacier data set.

Of course the values of Cs and $\tau$ impact on the results. It seems to us that the reviewer is disturbed by the conclusions of possible survival of the valley glaciers on the Italian Alps.

Missing error estimation: Because the model is not calibrated, there is no model error reported. The uncertainties given in Fig. 8 are induced by the different CORDEX ensemble members, but not by the model. Robust scientific results rely on a rigorous error estimation, would have helped to phrase a stronger discussion section.

We do not see the relationship between calibration and model error. The model error can be evaluated only thought the uncertainty of Cs, $\tau$ and temperature.

References

Farinotti, D., Brinkerhoff, D. J., Clarke, G. K. C., Fürst, J. J., Frey, H., Gantayat, P., Gillet-Chaulet, F., Girard, C., Huss, M., Leclercq, P. W., Linsbauer, A., Machguth, H., Martin, C., Maussion, F., Morlighem, M., Mosbeux, C., Pandit, A., Portmann, A., Rabatel, A., Ramsankaran, R., Reerink, T. J., Sanchez, O., Stentoft, P. A., Singh Kumari, S., van Pelt, W. J. J., Anderson, B., Benham, T., Binder, D., Dowdeswell, J. A., Fischer, A., Helfricht, K., Kutuzov, S., Lavrentiev, I., McNabb, R., Gudmundsson, G. H., Li, H. and Andreassen, L. M.: How accurate are estimates of glacier ice thickness? Results from ITMIX, the Ice Thickness Models Intercomparison eXperiment, The Cryosphere, 11(2), 949–970, doi:10.5194/tc-11-949-2017, 2017.

Leclercq, P. W., Oerlemans, J.: Global and hemispheric temperature reconstruction from glacier length fluctuations, Clim Dynam, 38(5-6), 1065-1079, https://doi.org/10.1007/s00382-011-1145-7, 2012.

Maussion, F., Butenko, A., Champollion, N., Dusch, M., Eis, J., Fourteau, K., Gregor, P., Jarosch, A. H., Landmann, J., Oesterle, F., Recinos, B., Rothenpieler, T., Vlug, A., Wild, C. T. and Marzeion, B.: The Open Global Glacier Model (OGGM) v1.1, Geoscientific Model Development, 12(3), 909–931, doi:10.5194/gmd-12-909-2019, 2019.

Oerlemans, J.: Glaciers and Climate Change, A.A Balkema, Lisse., 2001.

Oerlemans, J.: Extracting a climate signal from 169 glacier records., Science, 308(5722), 675–677, doi:10.1126/science.1107046, 2005.

Serandrei-Barbero, R., Donnici, S., and Zecchetto, S.: Projected effects of temperature changes on the Italian Western Tauri glaciers (Eastern Alps), J Glaciol, 1-10, https://doi.org/10.1017/jog.2019.7, 2019.

Zecchetto, S., Serandrei-Barbero, R. and Donnici, S.: Temperature reconstruction from the length fluctuations of small glaciers in the eastern Alps (northeastern Italy), Climate Dynamics, 49(1–2), 363–374, doi:10.1007/s00382-016-3347-5, 2017.